# Local measures of entanglement in black holes and CFTs

**Andrew Rolph**

Institute for Theoretical Physics, University of Amsterdam,
1090 GL Amsterdam, The Netherlands
Martin A. Fisher School of Physics, Brandeis University, Waltham, MA 02453, USA

★ andrew.d.rolph@gmail.com

## Abstract

We study the structure and dynamics of entanglement in CFTs and black holes. We use a local entanglement measure, the entanglement contour, which is a spatial density function for von Neumann entropy with some additional properties. The entanglement contour can be calculated in many 1+1d condensed matter systems and simple models of black hole evaporation. We calculate the entanglement contour of a state excited by a splitting quench, and find universal results for the entanglement contours of low energy non-equilibrium states in 2d CFTs. We also calculate the contour of a non-gravitational bath coupled to an extremal AdS$_2$ black hole, and find that the contour only has finite support within the bath, due to an island phase transition. The particular entanglement contour proposal we use quantifies how well the bath's state can be reconstructed from its marginals, through its connection to conditional mutual information, and the vanishing contour is a reflection of the protection of bulk island regions against erasures of the boundary state.



# 1 Introduction

Entanglement is an important topic in holography, condensed matter physics, and quantum information. In entangled states it is possible to be certain about the state of the full system and yet uncertain about the state of a subsystem, and a commonly used measure of this uncertainty is the von Neumann entropy, which is defined in terms of the density matrix $\rho$:

$$S := -\operatorname{Tr} \rho \log \rho \,. \tag{1}$$

For a pure state on a bipartite system $A \cup \bar{A}$, the von Neumann entropy of the reduced state $\rho_A := \operatorname{Tr}_{\bar{A}} \rho$ is non-zero if and only if $A$ and $\bar{A}$ are entangled.

Entanglement is nonlocal, so does it make sense to consider local measures of it? Entanglement may be nonlocal yet the interactions that govern the preparation and evolution of entangled states are (often) taken to be local. We want to study and apply measures of entanglement that are sensitive to the local physics. Von Neumann entropy when applied to spatial subregions is only a quasi-local measure of entanglement, and is insensitive to the structure and dynamics of entanglement between local degrees of freedom within the subregion. The entropy quantifies the total entanglement between the subregion and its purifier, and nothing else. These considerations motivate finding local entanglement measures.

One such measure is the entanglement contour, first introduced in [1], which can be thought of as a spatial density function for von Neumann entropy. The contour quantifies how much different degrees of freedom within a given subregion contribute to its von Neumann entropy. It is reasonable to suppose that some contribute more than others; in finite energy states of local field theories the von Neumman entropy of any subregion is UV divergent [2, 3], and this comes from the entanglement of degrees of freedom near the subregion's boundary, and so in some sense these degrees of freedom contribute the most to the entropy.

There have been several proposed formulas for the entanglement contour [1, 4, 5]. The particular proposal we will use originates from [4]. When $A$ is an interval $[x_1, x_2]$ on a line, which all our applications will be, the entanglement contour proposal is well-defined and

equals

$$s_A(x) = \frac{1}{2}\frac{d}{dx}\left(S([x_1, x]) - S([x, x_2])\right),\qquad(2)$$

with domain $x \in A$. One can check that integrating this contour over $A$ gives the von Neumann entropy. Less obviously it is also non-negative. These are just two of the seven properties this proposal satisfies [5]. One objection to the formula (2) is that mathematically it is a mere repackaging of the entanglement entropy of various subregions, but the same objection can be levelled at quantities generally accepted to be useful such as mutual information and conditional entropy. The value is in the physical interpretation.

The requirement that the entanglement contour be a density function for the von Neumann entropy far from uniquely defines it. The approach taken in the literature has been to reduce the space of possible formulas by specifying additional physically-motivated properties for the entanglement contour to satisfy, with the hope that with enough sufficiently constraining requirements the entanglement contour will be uniquely defined. This search for uniqueness is undermined somewhat by the fact that in holographic theories the boundary flux density of a flux-maximising bit thread configuration is a density function for the boundary subregion's von Neumann entropy, so is a natural entanglement contour candidate, and yet those thread configurations and their boundary flux densities are highly non-unique [6–9]. In examples where there exists a special set of bit threads based on geodesics [10] the boundary bit thread flux density has been calculated and shown to equal the entanglement contour calculated with the formula used in this paper [7, 11].

We are agnostic as to whether (2) is the 'true' and unique entanglement contour, or indeed whether there ought to be a unique contour. The proposal is equal to a certain conditional mutual information and so, even if one completely dismisses the notion of entanglement contours, our results are of independent interest. Furthermore, supposing that there ought to be a unique entanglement contour, the proposal (2) is an excellent candidate because it uniquely satisfies a certain set of physical requirements in 2d relativistic theories, and agrees with other proposals in their overlapping regimes of applicability [5]. Other points in the proposal's favour are that it is the most broadly applicable and tractable of existing entanglement contour proposals, and that it gives physically reasonable entanglement contours in known examples. The proposal's main weakness is that it is only well-defined in 1+1d. In higher dimensions there are a few finely-tuned examples with sufficient symmetry, such as when $A$ is a ball or infinite strip in the Minkowski plane, that the entanglement contour is effectively one-dimensional which makes the contour well-defined and calculable [5]. In the general non-symmetric higher dimensional case the formula (2) does not straightforwardly generalise.

Our first application is to entanglement dynamics. Entanglement dynamics of out-of-equilibrium systems is an area where von Neumann entropy has been a valuable tool [12–15], however there are perhaps some cases where a more local measure of entanglement would be pertinent. Such examples include splitting quenches [12, 16], where two halves of a system are decoupled from each other, as the von Neumann entropy of either half is constant under post-quench Hamiltonian evolution and so tells us nothing about the post-quench entanglement dynamics. This motivates the use of other, more local entanglement measures. We will calculate the entanglement contour of a 2d massless Dirac fermion CFT after a splitting quench, and also find universal results and dynamics for the entanglement contours of low energy excited states in 2d CFTs.

Our second application is to the study of the entanglement structure of Hawking radiation. In holographic systems, the Ryu-Takayanagi prescription [17] and its generalisations [18–20] have, among other things, been behind recent progress in resolving the black hole information paradox [21, 22]. For a black hole formed from collapse of matter in a pure state, unitarity requires the late Hawking radiation to be entangled with and purify the earlier radiation [23].

We are now able to calculate Page curves consistent with unitarity, and yet there is still relatively little known about *how* the late radiation purifies the early radiation, i.e. the precise microscopic details of the structure of entanglement between the late and early radiation. The von Neumann entropy, which reduces all the information contained in a density matrix to one number, is perhaps too blunt a tool for this purpose, which again motivates the consideration of other, more local measures.

We can calculate the entanglement contour of radiation in any 2d model of black hole evaporation, if the entanglement entropies of the intervals used in the contour formula are known. We explicitly calculate entanglement contour of a non-gravitational bath coupled to a zero temperature $AdS_2$ black hole, which is one of the simplest setups where islands appear [24]. We find that the bath's entanglement contour approximately vanishes everywhere except for a finite interval next to the 'black hole' degrees of freedom at the end of the half-line, and that this is caused by an island phase transition. This is because the entanglement contour quantifies how well the bath's state can be reconstructed from its marginals, through its connection to conditional mutual information. Our calculation is only a first step in using local measures of entanglement to probe Hawking radiation, because the setup is static and so not a good model for black hole evaporation. We leave entanglement contours of finite temperature and dynamic black holes to future work.

In section 2 we review how entanglement contours and partial entanglement entropy are abstractly defined and then in section 3 explore in detail the particular proposal we will use in the rest of the paper. In section 4 we apply this proposal to study the entanglement dynamics of out-of-equilibrium states in two setups: one after a splitting quench, and one a general weakly excited CFT state. In section 5 we calculate the entanglement contour of a non-gravitational bath coupled to a black hole using the island formula. In section 6 we discuss obstacles to generalising the proposal to higher dimensions. In section 7 we conclude with ideas about possible future research.

## 2 Review of entanglement contours

### 2.1 Entanglement contours and partial entanglement entropy

The entanglement contour $s_A(x)$, first introduced in [25], can be thought of as a density function for the von Neumann entropy[1]

$$S(A) := -\operatorname{Tr} \rho_A \log \rho_A, \tag{3}$$

for the reduced density matrix on $A$. There are several proposals for how precisely to define $s_A(x)$, which we will discuss soon, but for now it suffices to say that as a density function any entanglement contour proposal should obey the basic normalisation requirement

$$\int_A s_A(x) = S(A). \tag{4}$$

Given a partition of $A$ into $n$ disjoint subregions $A = A_1 \cup ... \cup A_n$, we also have an extensive quantity $s_A(A_i)$ called the partial entanglement entropy (PEE). It quantifies how much the degrees of freedom in $A_i \subseteq A$ contribute to $S(A)$. Like the entanglement contour it also obeys a normalisation requirement,

$$\sum_i^n s_A(A_i) = S(A). \tag{5}$$

---

[1]Density functions for von Neumann entropy (i.e. entanglement contours) should not to be confused with entanglement densities as introduced in [26].

The PEE and entanglement contour are related, in that we can calculate one from the other. The partial entanglement entropy equals the entanglement contour integrated over a subregion $A_i \subseteq A$,

$$s_A(A_i) = \int_{A_i} s_A(x). \tag{6}$$

Conversely, given a prescription for calculating the partial entanglement entropies for an arbitrary partition of $A$, we can define the entanglement contour as the infinitesimal limit

$$s_A(x) = \lim_{|A_i| \to 0} \frac{s_A(A_i)}{|A_i|}, \tag{7}$$

with $A_i \in \{A_i \subset A | x \in A_i\}$. This limiting procedure is not always unambiguous, especially in higher dimensions where the limit could depend on the shape of $A_i$ as its volume $|A_i|$ is shrunk to zero. Moreover, while the partial entanglement entropy makes sense as a quantity in both discrete and continuous systems, the entanglement contour can only be defined in continuum theories. For these reasons it is common to start with a partial entanglement entropy proposal, and from it define an entanglement contour function if and when it makes sense to do so. Our calculations will only deal with continuum theories on a line, for which the limiting procedure (7) defining the contour is happily unambiguous, so we can freely go back and forth between PEE and entanglement contour.

The progression from von Neumann entropy to partial entanglement entropy to entanglement contour can be thought of as a series of progressively more spatially fine-grained and local entanglement measures. This progression $S(A) \to \{s_A(A_1), ..., s_A(A_n)\} \to s_A(x)$ partitions $A$ into smaller and smaller subdivisions.

In our notation $s_A$ is unfortunately an overloaded symbol, though from its argument it will always be unambiguous whether $s_A$ is the PEE or the entanglement contour. $S(A)$ is the von Neumann entropy of $A$, $s_A(A_i)$ the partial entanglement entropy of $A_i \subset A$, and $s_A(x)$ the entanglement contour at $x \in A$.

## 2.2 Required properties

The normalisation requirement (4) is not particularly constraining on what the entanglement contour can be. To remedy this, previous literature has introduced sets of additional 'physical' requirements in an attempt to reduce the space of allowed entanglement contour proposals enough to make it unique. Some requirements that have been proposed include (for an arbitrary partition of $A$ into $\{A_1, ..., A_n\}$):

(I.) **Normalisation**: $\sum_{i=1}^{n} s_A(A_i) = S(A)$.

(II.) **Additivity**: $s_A(A_i \cup A_j) = s_A(A_i) + s_A(A_j)$.

(III.) **Non-negativity**: $s_A(A_i) \geq 0$.

(IV.) **Upper bound**: $s_A(A_i) \leq S(A_i)$.

(V.) **Invariance under local unitaries**: $s_A(A_i)$ is invariant under $\rho_A \to U_A \rho_A U_A^\dagger$ if $U_A = U_{A_i} \otimes U_{A \setminus A_i}$.

(VI.) **Spatial symmetry**: Symmetries of the state are symmetries of the entanglement contour. $U_T \rho_A U_T^\dagger = \rho_A$ and $T : A_i \leftrightarrow A_j \implies s_A(A_i) = s_A(A_j)$.

(VII.) **Permutation symmetry**: The formula for $s_A(A_i)$ should be invariant under the exchange of $A_i$ and $\bar{A}$.

Requirements (I.) and (II.) are certainly the most fundamental of the seven listed here. Without them one cannot have a correctly normalised entanglement contour. The additivity property is something that distinguishes partial entanglement entropy from most other quantum information quantities. Mutual information is not a good partial entanglement entropy candidate because it is not an extensive quantity, except for special cases such as the 2d free massless fermion CFT [27, 28]. Compared to (I.) and (II.), requirements (III.)-(VII.) are on less solid ground, and the question of which to impose, or if there are additional physical properties the partial entanglement entropy should satisfy that have not yet been thought of, is unresolved.

Many of the requirements follow from supposing that $s_A(A_i)$ is a measure of correlation between $A_i$ and $\bar{A}$, with $\bar{A}$ the region that purifies the state on $A$. Requirement (III.) is motivated by saying that $A_i$ is either correlated with $\bar{A}$ or it is not, and if $s_A(A_i)$ is a measure of correlation between them then it does not make sense for it to negative[2]. On the other hand though, in holographic theories the boundary flux density of a flux maximising bit thread configuration acts as a entanglement entropy density function for the boundary subregion in question, and yet that flux density is both highly non-unique and does not need to be positive. Requirement (IV.) also follows from the supposition that $s_A(A_i)$ is some kind of measure of correlation between $A_i$ and $\bar{A}$; it says that the correlation between $A_i$ and $\bar{A}$ should not be larger than the correlation between $A_i$ and the region complementary to $A_i$, which is a superset of $\bar{A}$. (V.) follows from the notion that local unitaries should not affect the total entanglement between $A_i$ and $\bar{A}$, and (VII.) says that a measure of the correlation between $A_i$ and $\bar{A}$ should be invariant under switching them.

All these requirements on the partial entanglement entropy can be converted to requirements on the entanglement contour by taking the infinitesimal limit through the contour's definition (7). For example, the normalisation requirement translates to

$$\int_A dx\, s_A(x) = S(A)\,. \tag{8}$$

It is worth noting that the trivial 'flat' entanglement contour $s_A(x) = S(A)/|A|$, which is undesirable as an entanglement contour as it contains zero information about the spatial structure of the entanglement, actually manages to satisfy every requirement except (IV.) and (VII.) for any state and theory[3].

There have been several proposed formulae for partial entanglement entropy, each with different regimes of validity. The first proposal, which we will not use, is applicable to free fermionic lattices [1]. Its extension to harmonic lattices is valid for Gaussian states, and defined in terms of the correlation matrix that characterises Gaussian states [29]. An $AdS_3/CFT_2$ holographic proposal was given in [4] that prescribes the partial entanglement entropy of a subinterval $A_i \subset A$ to be the length of a segment of the Ryu-Takayanagi surface associated by bulk modular flow, but is well-defined only when the bulk modular Hamiltonian of the region enclosed by the Ryu-Takayanagi surface is local.

In order to calculate entanglement contours we are forced to adopt a proposed formula. The one we choose is tractable, satisfies a physically reasonable set of requirements, and gives entanglement contours in known examples that are not obviously nonsensical. It has also been claimed that requirements (I.)-(VII.) are sufficiently constraining to rule out all but this one entanglement contour function, at least in Poincaré invariant theories [5]. We do not rule out the possibility of alternative requirements than the seven listed, or alternative proposals, though any list of requirements should at least be sufficiently constraining on allowed proposals such that they share the same rough qualitative features, otherwise the proposals are physically meaningless.

---

[2]We thank Qiang Wen for discussion on this point.

[3]Requirement (IV.) becomes a trivial bound for finite energy states in continuum field theories, where the entanglement entropy of any region is UV divergent.

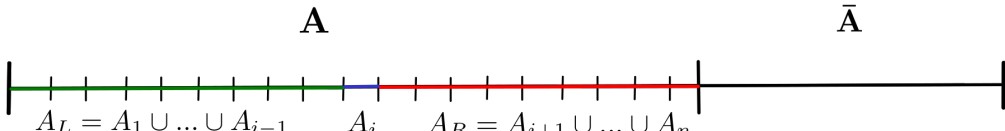

Figure 1: A bipartite system, with subregion $A$ and its complement $\bar{A}$, and with $A$ partitioned into $n$ subintervals. Partial entanglement entropy divides up the von Neumann entropy of $A$ amongst each of these subintervals. The entanglement contour is roughly speaking the $n \to \infty$ limit.

# 3 The conditional mutual information (CMI) proposal for entanglement contours

We start by reviewing the definition and motivation of the proposal we will be applying in calculations, then we will explore its relation to conditional mutual information, its properties, and its connection to kinematic space.

## 3.1 Review: Definition and motivation of CMI proposal

The partial entanglement entropy proposal we will use is unambiguously defined when there is a natural ordering to the subregions in the partition of $A$ (which is generally the case in 1+1d but not higher dimensions, though with some exceptions as we will discuss in Sec. 6). When $A$ is an interval partitioned into a totally ordered set of subintervals $\{A_1, ..., A_n\}$, the proposal is that the partial entanglement entropy of each subinterval is

$$s_A(A_i) = \frac{1}{2}(S(A_L A_i) - S(A_L) + S(A_i A_R) - S(A_R)), \tag{9}$$

where $A_L := A_1 \cup ... \cup A_{i-1}$ and $A_R := A_{i+1} \cup ... \cup A_n$ (see Fig. 1).

The formula (9) will be our workhorse, so given its importance it is worthwhile to explain its origin. This proposal for partial entanglement entropy was first introduced in [4] for $n = 3$ and generalised in [7]. The proposal is based on a particular bijective map along so-called modular planes in $\text{AdS}_3/\text{CFT}_2$ between points on the Ryu-Takayanagi surface and the boundary subregion $A$ it is homologous to, and then defining the partial entanglement entropy of a subdivision of $A$ as the length of the segment of the RT surface that the subdivision maps to [4]. While the modular plane construction is limited to cases where there is a local modular flow, the partial entanglement entropy formula (9) is not. The CMI proposal has also been claimed to be the unique formula for partial entanglement entropy in Poincaré invariant systems satisfying the seven physical requirements listed earlier [5].

Other calculations of entanglement contours using the CMI proposal have included: $\text{CFT}_2$ thermal states [4]; holographic warped CFTs [30]; defect CFTs, CFTs with mass deformations, states after global and local Cardy-Calabrese quenches, and heavy operator insertions [7]; holographic states dual to Bañados geometries, and general excited states in the small interval limit [31]. Near completion of this paper we became aware of [32] which also calculates entanglement contours in holographic boundary CFT models of black holes, which are different but similar to the model we will use, and also finds that the contour discontinuously vanishes due to an island phase transition.

To avoid confusion, note that in Sec. 2 '$s_A$' denoted partial entanglement entropy, but in an abstract sense as some loosely defined quantity that satisfies some set of required properties. From this point onwards '$s_A$' denotes the specific CMI proposal for partial entanglement

entropy, and the entanglement contour which is calculated from it, rather than something abstract.

## 3.2 Relation to conditional mutual information

We call (9) the CMI proposal, because it can be equivalently written as a conditional mutual information

$$s_A(A_i) = \frac{1}{2}I(A_i : \bar{A}|A_L) = \frac{1}{2}I(A_i : \bar{A}|A_R), \tag{10}$$

if the state on $A \cup \bar{A}$ is pure[4].

Why is conditional mutual information a natural proposal for partial entanglement entropy? Given a partitioning of $A$, the partial entanglement entropy is supposed to quantify how much each subset $A_i$ contributes to the von Neumann entropy of $A$, which is a measure of entanglement between $A$ and a purifying system $\bar{A}$, with that entanglement quantified by mutual information

$$S(A) = \frac{1}{2}I(A : \bar{A}). \tag{11}$$

Now we imagine building up the correlation between $A = A_1 \cup ... \cup A_n$ and the purifying system $\bar{A}$ piece by piece. First we build up the correlation between $A_1$ and $\bar{A}$, then between $A_2$ and $\bar{A}$ but conditioned on $A_1$, and so on. The amount of correlation is quantified by the mutual information, and this step by step building up of correlation is what underlies the chain rule for conditional mutual information:

$$I(A_1...A_n : \bar{A}) = I(A_1 : \bar{A}) + I(A_2 : \bar{A}|A_1) + ... + I(A_n : \bar{A}|A_1...A_{n-1}). \tag{12}$$

Since this represents the building up of correlation between $A$ and $\bar{A}$ piece by piece, it is natural to interpret each term in this sum as the partial entanglement entropy of the corresponding subinterval:

$$s_A(A_i) = \frac{1}{2}I(A_i : \bar{A}|A_L), \tag{13}$$

where $A_L := A_1 \cup ... \cup A_{i-1}$. Hence the name 'CMI proposal'. The factor of $1/2$ relative to the terms in (12) comes from the $1/2$ in (11).

We built up the correlation between $A$ and $\bar{A}$ piece by piece going from left to right, but why not right to left? It does not make a difference, we get exactly the same proposal because of the duality relation of conditional mutual information,

$$I(A_i : \bar{A}|A_L) = I(A_i : \bar{A}|A_R), \tag{14}$$

where $A_R := A_{i+1} \cup ... \cup A_n$. This duality relation is straightforwardly checked using the definition of conditional mutual information

$$I(A : B|C) := S(AC) + S(BC) - S(ABC) - S(C). \tag{15}$$

Using this definition for conditional mutual information, we can also write the CMI proposal just in terms of von Neumann entropies of subregions of $A$

$$s_A(A_i) = \frac{1}{2}(S(A_L A_i) - S(A_L) + S(A_i A_R) - S(A_R)), \tag{16}$$

as in (9). Writing the proposal this way makes it clear that the CMI proposal gives partial entanglement entropies that depend only on the state on $A$, and not how the state is purified by degrees of freedom in $\bar{A}$.

---

[4]We may assume the state on $A \cup \bar{A}$ to be pure without loss of generality, because if the state is mixed then we are free to add auxiliary degrees of freedom to $\bar{A}$ to purify the state, then redefine that as the new $\bar{A}$, and it turns out that for the CMI proposal it does not actually matter what $\bar{A}$ is, only that it purifies the state on $A$.

### 3.3 From partial entanglement entropy to entanglement contour

Taking $A$ to be an arbitrary interval $[x_1, x_2]$, the CMI proposal for entanglement contours follows from (9) by taking the number of subintervals $n \to \infty$ and their size to zero, to get:

$$s_A(x) = \frac{1}{2}\frac{d}{dx}\left(S([x_1, x]) - S([x, x_2])\right). \tag{17}$$

The entanglement contour is not defined for $x$ outside of $A$.

There are already a number of results in the literature for entanglement contours calculated using the CMI proposal, to which we will add two new ones. The simplest contour that can be calculated is that of an interval $A = [x_1, x_2]$ for the vacuum state of a CFT on $\mathbb{R}^{1,1}$:

$$s_A(x) = \frac{c}{6}\frac{x_2 - x_1}{(x - x_1)(x_2 - x)}, \tag{18}$$

with $x \in [x_1, x_2]$. This result follows from plugging into the contour formula (17) the single interval CFT vacuum entanglement entropy [33]

$$S([x_1, x_2]) = \frac{c}{3}\log\left(\frac{x_1 - x_2}{\epsilon}\right). \tag{19}$$

This vacuum entanglement contour (18) is finite except where it diverges near the edges of $A$, and reaches a minimum at the interval's midpoint, though still order $c$. Two intuitions about CFT vacuum states that the entanglement contour supports are that (1) UV divergences in the von Neumann entropy of a subregion come from short range entanglement across the subregion's boundary, and (2) in gapless systems degrees of freedom away from the boundary can contribute substantially to the entanglement entropy.

### 3.4 Properties

Some properties of the CMI proposal (13) follow immediately from its definition in terms of conditional mutual information:

- The sum $\sum_i s_A(A_i)$ is correctly normalised to $S(A)$, as follows directly from the chain rule (12).

- The CMI proposal always gives non-negative partial entanglement entropies, because conditional mutual information is always non-negative, which is equivalent to the strong subadditivity relation of von Neumann entropies.

- As mentioned earlier the CMI proposal has a $L \leftrightarrow R$ symmetry which is not immediately obvious when written in the form (16), but nonetheless holds,

$$s_A(A_i) = \frac{1}{2}I(A_i : \bar{A}|A_L) = \frac{1}{2}I(A_i : \bar{A}|A_R). \tag{20}$$

- $s_A(A_i)$ is finite, except when $A_i$ shares a border with $A$. The individual entropy terms in the definition (16) are UV divergent for finite energy states in local field theories, but these divergences are (theory-dependent) functions of the boundary geometry [2],

$$S(V) = g_{d-1}[\partial V]\epsilon^{-(d-1)} + ... + g_0[\partial V]\log\epsilon + S_{finite}(V), \tag{21}$$

and the combination of terms in (16) is such that the boundary divergences cancel and make $s_A(A_i)$ finite, except when $A_i$ shares a border with $A$.

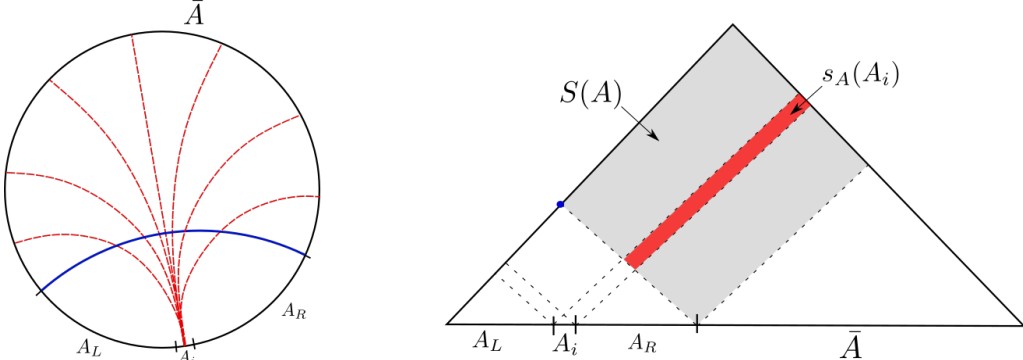

Figure 2: Left: A time slice of an asymptotically AdS$_3$ geometry. The blue curve is the RT surface of boundary region $A$, and the red curves are representatives of the set of all geodesics that connect $A_i$ to $\bar{A}$. Right: Kinematic space, where each point represents a boundary-anchored geodesic in AdS$_3$. The blue dot is the RT surface. The grey region is the set of all geodesics between $A$ and $\bar{A}$, whose volume is $S(A)$. The red region is the set of geodesics between $A_i$ and $\bar{A}$, whose volume is the CMI proposal for partial entanglement entropy.

- When $s_A(A_i)$ is small there cannot be much entanglement between $A_i$ and the system that purifies $A$. This is because in holographic theories conditional mutual information upper bounds mutual information, and this is called monogamy of mutual information, while in non-holographic theories it upper bounds the squashed entanglement (see appendix A for more details, references, and other relevant conditional mutual information properties).

- $s_A(A_i)$ does not depend on how the rest of $A\backslash A_i$ is partitioned. This is because 'links' in the chain rule (12) can be freely split apart and joined together without affecting the CMI of 'links' elsewhere along the chain, as can be seen in the independence of (13) from how $A_L$ is subdivided. If we combine subdivisions $A_i$ and $A_{i+1}$ into a single subdivision, so defining a new partitioning of $A$, we have the additivity property:

$$s_A(A_i) + s_A(A_{i+1}) = s_A(A_i \cup A_{i+1}), \tag{22}$$

and the partial entanglement entropies of other subdivisions are unaffected.

The CMI proposal is ambiguously defined in higher dimensions. When there is only one spatial dimension, there is a natural ordering to the subdivisions of $A$ inherited from their left to right ordering on the line, but not so in higher dimensions. There can also be ambiguities in ordering when $A$ is the union of disjoint regions, even in 1+1d. These issues and possible resolutions will be discussed in greater depth in section 6. In our applications we will only consider entanglement contours of single intervals in 1+1d, where the CMI proposal is unambiguously defined.

## 3.5 Relation to kinematic space

In AdS$_3$/CFT$_2$, the CMI proposal has a simple and natural interpretation in kinematic space. $s_A(A_i)$ equals the volume of the region of kinematic space that contains geodesics with one endpoint on $A_i$ and one on $\bar{A}$ (see Fig. 2). It is not surprising that the CMI proposal is better understood in kinematic space than position space, since the volume form of kinematic space is a conditional mutual information [34,35]. The CMI proposal can be understood as one way

of dividing up the region of kinematic space whose volume is $S(A)$ amongst the subregions in the partition of $A$.

To review kinematic space, it is the 2d space of boundary-anchored oriented geodesics on a time slice of time reflection symmetric asymptotically AdS$_3$ geometry. The Crofton formula equates the length of a curve $\gamma$ in position space with the volume of the region in kinematic space $K$ containing all geodesics that intersect $\gamma$,

$$\frac{\text{Length}\,(\gamma)}{4G_N} = \frac{1}{4}\int_K \omega(u,v)n_\gamma(u,v)\,. \tag{23}$$

Coordinates $(u,v)$ parametrise the two boundary endpoints of a single geodesic, $n_\gamma(u,v)$ is the intersection number of that geodesic with $\gamma$, and $\omega(u,v)$ is the Crofton volume form which is given by a conditional mutual information involving three intervals:

$$\begin{aligned}\omega(u,v) &= \frac{\partial^2 S(u,v)}{\partial u \partial v}dudv \\ &= I([u-du,u]:[v,v+dv]|[u,v])\,.\end{aligned} \tag{24}$$

The von Neumann entropy $S(A)$ is the volume of a certain causal diamond $\blacklozenge_A$ in kinematic space depicted by the grey region in Fig. 2. By definition, partial entanglement entropy apportions $S(A)$ amongst subdivisions of $A$. This can be done by constructing a bijective map $\zeta$ between subdivisions of $A$ and subdivisions of the region $\blacklozenge_A$ in kinematic space, as then the partial entanglement entropy $s_A(A_i)$ can be defined as the volume of the region of kinematic space volume $A_i$ maps to:

$$s_A(A_i) = \frac{1}{4}\text{Vol}\,(\zeta(A_i))\,. \tag{25}$$

$\zeta(A_i)$ is the red region on the right of Fig. 2. Any such bijective map, as a way of dividing up $S(A) = \text{Vol}(\blacklozenge_A)$, trivially satisfies $\sum_i s_A(A_i) = S(A)$.

From this perspective, the CMI proposal $s_A(A_i) = \frac{1}{2}I(A_i : \bar{A}|A_R)$ for partial entanglement entropy is special because it comes from a particularly natural bijective map. This map is between $A_i$ and the intersection in kinematic space of null rays from $A_i$ with $\blacklozenge_A$, which is the red region in the right subfigure of Fig. 2. This subregion of kinematic space is the set of oriented geodesics that have one endpoint in $A_i$ and one in the purifying region $\bar{A}$, as depicted in the left subfigure of Fig. 2.

The holographic dual of the CMI proposal is different in character from the holographic duals of other quantum information quantities. It is a volume of a region of kinematic space, but not the length or volume of anything in the bulk position space. This is different in character from the bulk duals of other quantum information quantities, such as the entanglement of purification and complexity, which are the volumes of some codimension-1 or 0 surfaces [36,37].

The idea of the modular plane construction method is to divide up the Ryu-Takayanagi surface (in AdS$_3$) into segments, and equate the lengths of those segments to the partial entanglement entropies of subdivisions of $A$ [4]. The CMI proposal was inspired by this construction, and yet the regions of kinematic space that the CMI proposal and the modular plane construction associate to a given element in the partition of $A$ are different. Both regions only contain geodesics with one endpoint in $A$ and one in $\bar{A}$, but the former only includes geodesics with one endpoint on $A_i \subset A$, while the latter only includes geodesics that pass through the RT surface segment specified by the modular plane construction. It is mysterious to us that the volumes of these different regions of kinematic space happen to be equal, at least in the examples calculated so far: vacuum AdS$_3$ and BTZ black holes [4,38].

# 4 Local entanglement dynamics

The entanglement contour, as a local measure of entanglement, is well suited for giving us a sharp, spatially fine-grained picture of entanglement dynamics. Previous work has applied entanglement contours to study entanglement dynamics, primarily of states after quantum quenches [1, 7]. In this section we will explore two new examples: states after splitting quenches, and general low energy non-equilibrium states.

## 4.1 Splitting quench

Entanglement contours can tell us more about entanglement dynamics than entanglement entropy, and this is especially clear following a splitting local quench [12, 16]. In a splitting quench the system is instantaneously cut in half, decoupling the Hamiltonian across the cut,

$$H_{L \cup R} \rightarrow \mathbb{1}_L \otimes H_R + H_L \otimes \mathbb{1}_R \,. \tag{26}$$

After the quench, both $S(L)$ and $S(R)$ are constant, as the reduced density matrices each evolve unitarily under the decoupled Hamiltonian, and so the entanglement entropy of either half tells us nothing about the post-quench entanglement dynamics. The entanglement contour, in contrast, is sensitive to the post-quench dynamics.

We take the system to live on $\mathbb{R}^{1,1}$, and the split to be at $x = 0$ and $t = 0$. From the definition of the CMI proposal, in order to calculate the entanglement contour of say the right half $R = [0, \infty)$ we need to know the entanglement entropy of arbitrary connected subintervals of the halves post-quench,

$$s_R(x) = \frac{1}{2} \frac{d}{dx} (S([0, x]) - S([x, \infty))) \,. \tag{27}$$

There are a couple of systems and states for which these entropies are known, the simplest of which is the 2d massless Dirac fermion CFT which before the split is in its vacuum state [16]. For this CFT initially in its vacuum state the contour of one of the half-lines after a splitting local quench at $t = 0$ is

$$s_R(x, t) = \begin{cases} \frac{1}{6} \frac{t}{t^2 - x^2} \,, & 0 \le x < t \\ \frac{1}{6} \frac{x}{x^2 - t^2} \,, & t \le x \end{cases} \,. \tag{28}$$

This contour is a propagating wave that is semi-localised around its singular peak at $x = t$. An instant after the splitting quench, before the system has had time to evolve, the contour is

$$\lim_{t \to 0^+} s_R(x, t) = \frac{1}{6x} \,. \tag{29}$$

This diverges at the split point at $x = 0$ because of the pre-quench vacuum state's UV entanglement between the two halves. The time evolution of the entanglement contour is consistent with the quasiparticle picture [12]; pre-quench most of the entanglement between the two halves are from UV degrees of freedom across $x = 0$ and, after the split decouples the two halves, their quasiparticles freely propagate away from the cut and carry the entanglement away with contour velocity $v_c = 1$.

The main point we wish to make here is that though the total entanglement between the left and right halves is constant after the split, how that entanglement is distributed amongst the degrees of freedom is dynamic, and the entanglement contour is sensitive to those dynamics even when the entanglement entropy is not.

### 4.2 Weakly excited CFT states

We can also use entanglement contours to explore the entanglement dynamics of the class of states which are low energy perturbations of the vacuum state. To do this we use the first law of entanglement entropy, which relates the first order change in entropy to the change in expectation value of the modular Hamiltonian $K_\rho := -\log\rho$,

$$S(\rho + \epsilon\delta\rho) = S(\rho) + \epsilon\text{Tr}(\delta\rho K_\rho) + \mathcal{O}(\epsilon^2),\tag{30}$$

and follows from positivity of relative entropy [39]. The CMI proposal requires the entanglement entropy for single intervals and nothing else, so we can calculate the entanglement contour of any perturbed state for which the modular Hamiltonian of the unperturbed state is known (see also [40] for an entanglement contour version of the first law of entanglement entropy).

Many of the known modular Hamiltonians are for states in 2d CFTs, and can be written as the integral of the stress tensor times a local weight [41]. We consider weak excitations of the vacuum state $|\Omega\rangle$ of a CFT on $\mathbb{R}^{1,1}$, whose modular Hamiltonian of a spatial interval $[x_1, x_2]$ on a constant $t$ slice is known, universal, and given by an integral of the energy density [42]:

$$K^{vac.}_{[x_1, x_2]} = \frac{2\pi}{x_2 - x_1} \int_{x_1}^{x_2} d\tilde{x}(\tilde{x} - x_1)(x_2 - \tilde{x})T_{tt}(\tilde{x}).\tag{31}$$

Using (31) and the first law of entanglement, we can calculate the entanglement contour of a single interval $x \in [0, L]$ of an excited state in a 2d CFT,

$$\begin{aligned}
s_A(x, t) &= s_A^{vac.}(x, t) + \frac{1}{2}\frac{d}{dx}\left(\left\langle K^{vac.}_{[0,x]}\right\rangle - \left\langle K^{vac.}_{[x,L]}\right\rangle\right) + \dots \\
&\approx \frac{c}{6}\frac{L}{x(L-x)} + \pi\left[\int_0^x d\tilde{x}\frac{\tilde{x}^2}{x^2}\langle T_{tt}(\tilde{x}, t)\rangle + \int_x^L d\tilde{x}\frac{(\tilde{x}-L)^2}{(x-L)^2}\langle T_{tt}(\tilde{x}, t)\rangle\right].
\end{aligned}\tag{32}$$

We dropped terms quadratic and higher order in $L^2\langle T_{tt}\rangle \ll 1$. What this result gives us is a sharp picture of the local entanglement dynamics for any weakly excited 2d CFT state.

One interesting feature of this entanglement contour (32) is that we can determine the energy density at any point in the interval from knowledge of the entanglement contour in the point's neighbourhood:

$$2\pi L\langle T_{tt}(x, t)\rangle = \left[-x(L-x)\partial_x^2 - 3(L-2x)\partial_x + 6\right]\left(s_A(x, t) - s_A^{vac.}(x, t)\right).\tag{33}$$

This follows directly from (32). In contrast, knowledge of the entanglement entropy of a single interval in a 2d CFT only gives partial information about the energy density in that interval.

Now we specialise our general result (32) to the subclass of weakly excited CFT states whose energy density $\langle T_{tt}\rangle$ is localised, i.e. whose spatial support is much smaller than the interval width. We take a right-moving Gaussian wave of total energy $E$

$$\langle T_{tt}(x, t)\rangle = \frac{E}{\sqrt{\pi}\sigma}e^{-\frac{(x-t)^2}{\sigma^2}},\tag{34}$$

whose energy density is narrow ($\sigma \ll L$) and low ($E/\sigma \ll L^{-2}$). Tracelessness and conservation of the CFT stress tensor imply that $(\partial_t^2 - \partial_x^2)T_{tt}(x, t) = 0$, so that excitations propagate at the speed of light. For this energy density profile the entanglement contour is approximately

$$s_A(x, t) \approx \frac{c}{6}\frac{L}{x(L-x)} + \pi\left(\frac{t^2}{x^2}\Theta(x-t) + \left(\frac{t-L}{x-L}\right)^2\Theta(t-x)\right)E,\tag{35}$$

as plotted in Fig. 3. It is worth stating again that the entanglement contour $s_A(x)$ by definition can only have support on the spatial region $A$, which here is $x \in [0, L]$.

The vacuum-subtracted entanglement contour, given by the linear in $E$ term in (35), is a wave whose peak is centred on and travels with the localised energy density wavepacket at the speed of light. The contour also has a leading edge that travels ahead of the energy packet, and a trailing edge that relaxes to zero after the packet has left the interval.

The velocity of the entanglement contour depends on how the velocity is defined. If we define it as the velocity of the contour's peak, then for (35) we find an entanglement contour velocity of $v_c = 1$ for localised perturbations to the vacuum in 2d CFTs, as also found in [7]. If however we choose to define the contour velocity from the contour's mean position,

$$v_c = \frac{d}{dt}\langle x \rangle, \qquad \text{with} \quad \langle x \rangle := \frac{\int_0^L x\, s_A(x,t) dx}{\int_0^L s_A(x,t) dx}, \tag{36}$$

then we find a time dependent velocity

$$v_c = \frac{1}{2}\left( \left(\frac{L}{L-t}\right)^2 \log\left(\frac{L}{t}\right) + \left(\frac{L}{t}\right)^2 \log\left(\frac{L}{L-t}\right) - \frac{L^2}{t(L-t)} \right). \tag{37}$$

This contour velocity diverges when the peak of the contour is near the edges of the interval, and is at a minimum of $v_c \approx 0.77$ when the contour's peak reaches the interval's midpoint.

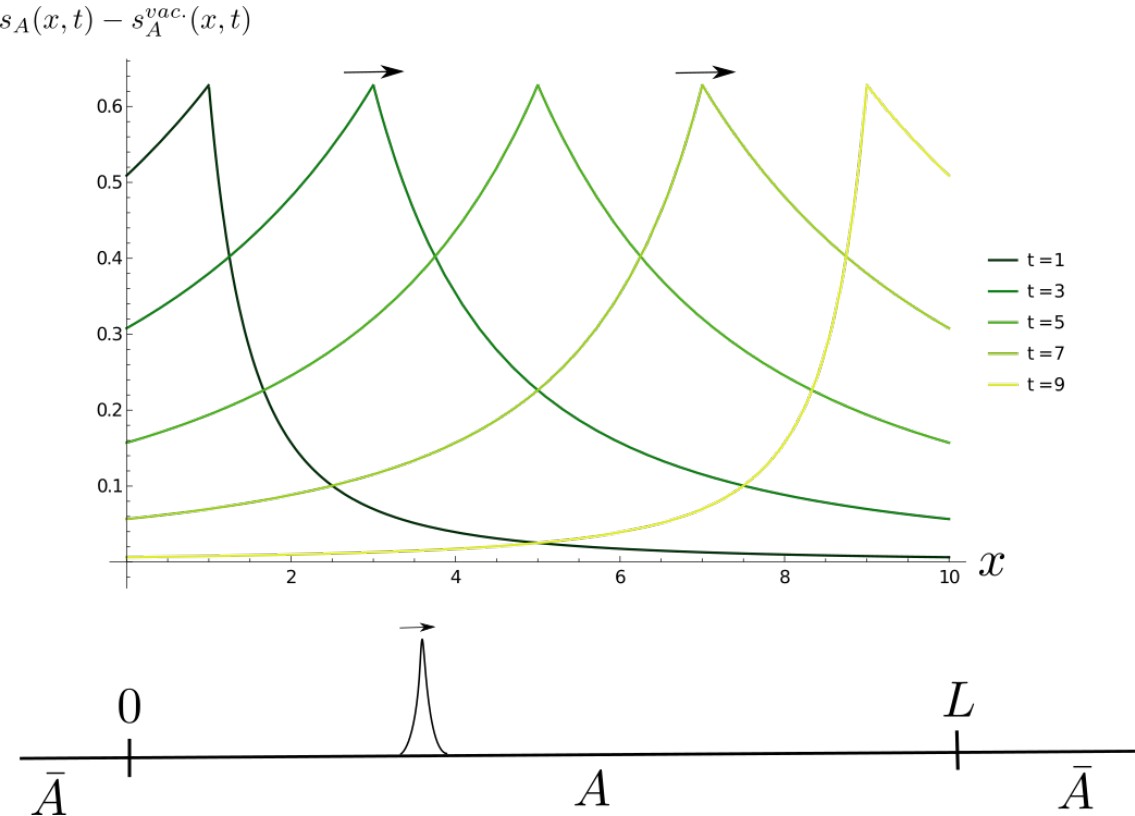

Figure 3: Lower figure: A single interval of a weakly excited 2d CFT on $\mathbb{R}^{1,1}$ with a localised low-energy pulse. Upper figure: Vacuum-subtracted entanglement contour of that interval, given by equation (35), and plotted with $E = 1/10$ and $L = 10$. The peak of the entanglement contour wave is centred on and travels with the energy pulse.

A superluminal entanglement contour velocity is not in tension with causality; entanglement is not a locally propagating object. Nonetheless, except for sharply localized contours, the contour velocity is not an unambiguously defined quantity, so the physical meaning we can ascribe to it is limited.

The entanglement contour given by (35) is a little strange from the quasiparticle perspective. If we imagine that the energy wave (34) acts as a source of pairs of quasiparticles, then the entanglement entropy of *A* is non zero when one of the quasiparticles is inside the interval and its partner outside, and the contour gives us the precise spatial distribution of partners inside the interval. The number of quasiparticles created roughly scales linearly with the total energy of the excitation. The fact that the leading edge of the contour moves ahead of the sharply localised energy wave suggests that the energy wave acts as a non-local source for quasiparticles over a distance of order *L*. On the other hand, for $t < 0$ the vacuum-subtracted entanglement contour is zero, so it would seem the energy wave only acts as a non-local source once it enters the interval and not before, which is difficult to interpret from a physical viewpoint as there is nothing physically different about the edge of the interval than any other point on the infinite line. This may suggest that the quasiparticle picture needs to be refined. It would be interesting to pursue this further, and to apply local measures of entanglement to get a sharper picture of entanglement dynamics in other systems.

## 5 Island contours

In this section we calculate the entanglement contour of a holographic system whose nongravitational description is a CFT with boundary degrees of freedom on a half-line, and whose gravitational description is JT gravity coupled to a flat space bath. Such systems have been useful as models of black hole evaporation, and their bath entanglement entropies give Page curves consistent with unitarity [21, 22, 43]. The motivation for applying a local measure of entanglement like the entanglement contour is in a sense to go beyond the Page curve, which is important, but only gives us the total entanglement between the black hole and Hawking radiation. We want a sharp, spatially fine-grained picture of the entanglement structure of Hawking radiation.

### 5.1 Review: Extremal AdS$_2$ black holes coupled to flat space baths

The black hole model we consider is perhaps the simplest with islands. It is an extremal AdS$_2$ black hole coupled to flat space bath, as studied in [24]. This system is static and so not a good model of black hole evapation, but we can learn about the impact of islands on the bath's entanglement structure, and leave dynamical evaporation to future work. We now review relevant details of the model.

JT gravity has the action[5]

$$I_{JT} = \frac{1}{4\pi} \int d^2x \sqrt{-g}[\phi R + 2(\phi - \phi_0)] + I_{bdy}, \tag{38}$$

whose equations of motion constrain the geometry to be locally AdS$_2$, and so whose solutions differ only in global structure. The action can be thought of as the dimensional reduction of the s-wave sector of the near-horizon region of a higher dimensional near-extremal black hole, with $\phi$ the volume of the transverse sphere. The particular solution we want is the extremal

---

[5]Taking $l_{AdS} = 1$ and absorbing $4G_N$ in a dilaton field redefinition

AdS$_2$ black hole, whose global geometry is the Poincaré patch of AdS$_2$

$$ds^2 = \frac{-dt^2 + dz^2}{z^2}, \tag{39}$$

and whose boundary cutoff we take to be at $z = -\epsilon$, so that $z \in (-\infty, -\epsilon]$. The dilaton solution

$$\phi = \phi_0 - \frac{\phi_r}{z}, \tag{40}$$

satisfies the equations of motion with boundary condition

$$\phi|_{z=-\epsilon} = \phi_0 + \frac{\phi_r}{\epsilon}. \tag{41}$$

We take this black hole and couple it to a non-gravitational flat half-line bath

$$ds^2 = -du^2 + d\sigma^2, \qquad \sigma \in [0, \infty), \tag{42}$$

by joining the end of the half-line bath at $\sigma = 0$ to the $AdS_2$ boundary at $z = -\epsilon$. Poincaré and boundary time are related by $t = \epsilon u$, and we extend the flat space coordinates to cover the AdS$_2$ region (39).

For the matter sector we add a 2d CFT with transparent conformal symmetry-preserving boundary conditions at the shared boundary. We take the matter CFT's central charge to be large, partly because we will need large-$c$ factorisation in our entropy calculations. The CFT lives half in the gravitational AdS$_2$ region and half in the non-gravitational flat space region. There are no joining quenches, backreactions, or radiation to worry about in this zero temperature eternally coupled model; indeed the only impact of gravity being dynamical in the AdS$_2$ region is that one should use the island formula when calculating entanglement entropy of regions in the non-gravitational bath.

We assume that our JT gravity plus matter system has a holographic quantum mechanical dual, so that the combined system of JT gravity coupled to a bath without dynamical gravity is holographically dual to a half-line CFT with boundary degrees of freedom at $\sigma = 0$, as shown in Fig. 4. We will distinguish intervals in the non-gravitational description from intervals in the gravitational description by using bold font for the former and non-bold font for the latter. We define the bath region $A$ to be the whole half-line system minus the degrees of freedom $\bar{A}$ at the end of the half-line, which is the open interval[6]

$$A := (\mathbf{0}, \mathbf{\infty}). \tag{43}$$

The bold font for the interval emphasises that we are calculating the entanglement contour in the *non-gravitational* description, by applying the island formula in the gravitational description. The degrees of freedom at the end of the half-line, $\bar{A}$, are the 'black hole' degrees of freedom because on their own their thermal states are holographically dual to AdS$_2$ black holes.

## 5.2 Calculation of the bath entanglement contour

What we want to calculate is the entanglement contour $s_A(x)$ of the bath. In order to calculate the contour, we further partition $A$ in two at the point $\sigma = x$,

$$A = A_L \cup A_R, \quad A_L := (\mathbf{0}, \mathbf{x}], \quad A_R := (\mathbf{x}, \mathbf{\infty}). \tag{44}$$

---

[6]We will be careful about whether an interval is open or closed, though it only really matters for intervals such as (43) or $A_L$ in (44), where open/closed intervals exclude/include the quantum mechanical system at the end of the half-line.

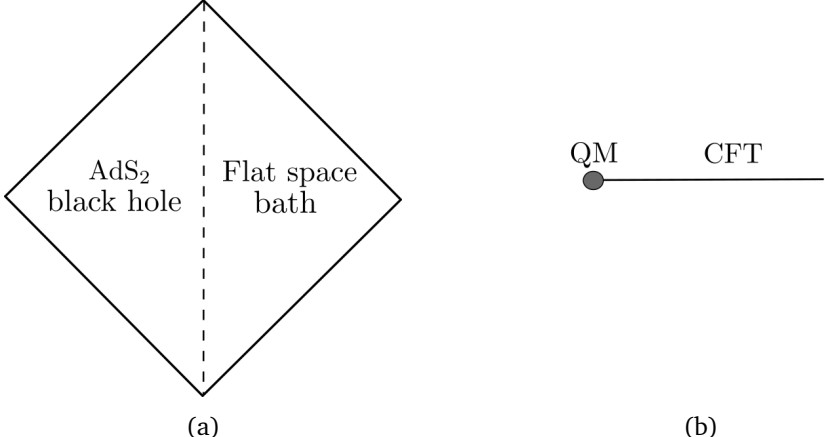

(a)                                                    (b)

Figure 4: (a) An extremal $AdS_2$ black hole coupled to a zero temperature flat space bath, with conformal matter. (b) The holographic dual, a CFT on a half line with 'black hole' degrees of freedom at the endpoint.

The entanglement contour at a point $x$ on the half-line is defined to be

$$s_A(x) = \frac{1}{2}\frac{d}{dx}(S(A_L) - S(A_R)). \tag{45}$$

We first calculate the entropy of $A_R$. To do so, we use the island formula, which gives the entropy of a spatial region in the non-gravitational description in terms of entropic and geometric quantities in the semiclassical gravitational description [43],

$$S([b,b']) = \min \underset{\mathcal{I}}{\mathrm{ext}} \left( S_{bulk}(\mathcal{I} \cup [b,b']) + \frac{\mathrm{Area}(\partial\mathcal{I})}{4G_N} \right), \tag{46}$$

where the minimisation is over the set of spatial regions that are extrema of the island functional on the right hand side. These extrema may include the trivial region which is the empty set.

In our JT gravity setup, the area term is given by the value of the dilaton at the endpoints of the interval. We assume we can restrict ourselves to considering single interval islands, parametrised by the positions of their endpoints $\mathcal{I} = [-a, -a']$. Then the island prescription tells us that the entropy is found by varying (46) with respect to those island endpoints,

$$S([b,b']) = \min \underset{a,a'}{\mathrm{ext}} \left( S_{bulk}([-a,-a'] \cup [b,b']) + \phi(-a) + \phi(-a') \right). \tag{47}$$

There are two special cases where the entropy of the union of two disjoint reduces to the entropy of a single interval: (1) when the island is trivial and empty, $\mathcal{I} = \varnothing$, which gives the no-island entropy

$$S_{no-island}([b,b']) = \frac{c}{3}\log\left(\frac{b'-b}{\delta}\right), \tag{48}$$

where $\delta$ is a UV regulator for the CFT, and (2) when $a = \infty$ and $b' = \infty$, because then we can use that the global state is pure and so the entanglement entropies of complementary regions are equal:

$$S_{bulk}([-\infty, -a'] \cup [b, \infty]) = S_{bulk}([-a', b])$$
$$= \frac{c}{6}\log\left(\frac{(a'+b)^2}{a'\delta^2}\right). \tag{49}$$

This has an extra $a'$ in the denominator compared to the standard flat space entanglement entropy result because one endpoint of the interval is in the AdS$_2$ region. The extra $a'$ comes from the Weyl transformation between AdS$_2$ and flat space, and can be thought of as accounting for the rescaling of the UV regulator at the interval's endpoint [21].

Besides these two special cases, the CFT vacuum entanglement entropy of the union of two disjoint intervals is given by a 4-pt function of twist operators and is not universal, so to progress we must make a further assumption about the matter CFT. We assume large-$c$ vacuum block dominance so that the entropy is approximately the sum of single interval entropies which scale linearly with $c$ [44],

$$S_{bulk}([-a,-a'] \cup [b,b']) = \begin{cases} S_{bulk}([-a,-a']) + S_{bulk}([b,b']) + O(c^0), & z < 1/2 \\ S_{bulk}([-a,b']) + S_{bulk}([-a',b]) + O(c^0), & z > 1/2 \end{cases}, \quad (50)$$

where $z$ is the conformal cross ratio

$$z = \frac{(a-a')(b'-b)}{(b+a)(b'+a')}. \quad (51)$$

When $z < 1/2$ and the s-channel dominates the 4-pt function of twist operators, the minimal extremum of the island prescription is the empty island. This is because having a non-trivial island can only increase the entropy, i.e. having $\mathcal{I} \neq \varnothing$ adds only non-negative contributions from the entropy term $S[-a,-a']$ in (50) and the dilaton in (47).

This means we only need to worry about non-trivial islands when $z > 1/2$, which is the t-channel phase. In this phase, using (50) and (47), we can extremise over $a$ and $a'$ independently, to give the island entropy:

$$\begin{aligned} S_{island}([\boldsymbol{b},\boldsymbol{b'}]) = &\min_{a} \operatorname*{ext} \left( \phi(-a) + S_{bulk}([-a,b']) \right) + \min_{a'} \operatorname*{ext} \left( \phi(-a') \right. \\ &\left. + S_{bulk}([-a',b]) \right) + O(c^0) \\ = &2\phi_0 + \frac{\phi_r}{a_{min}} + \frac{c}{6}\log\left(\frac{(a_{min}+b')^2}{a_{min}\delta^2}\right) + \frac{\phi_r}{a'_{min}} \\ &+ \frac{c}{6}\log\left(\frac{(a'_{min}+b)^2}{a'_{min}\delta^2}\right) + O(c^0), \end{aligned} \quad (52)$$

where $a_{min}$ is the values of $a$ at which the first line has the smaller of its two extrema with respect to $a$,

$$\begin{aligned} a_{min} &= \frac{1}{2}\left( b' + \frac{6\phi_r}{c} + \sqrt{b'^2 + 36\frac{\phi_r}{c}b' + 36\frac{\phi_r^2}{c^2}} \right) \\ &\approx \begin{cases} b', & b' \gg \frac{\phi_r}{c} \\ \frac{6\phi_r}{c}, & b' \ll \frac{\phi_r}{c} \end{cases}, \end{aligned} \quad (53)$$

and similarly for the minimising value of $a'$, meaning that $a'_{min}$ is given by (53) but with $b$ substituted for $b'$.

This gives our intermediate result towards calculating the contour, which is the entropy of a single interval of the half-line in the non-gravitational description:

$$S([\boldsymbol{b},\boldsymbol{b'}]) = \begin{cases} S_{island}([\boldsymbol{b},\boldsymbol{b'}]), & \text{if } z > 1/2 \text{ and } S_{island} < S_{no-island} \\ S_{no-island}([\boldsymbol{b},\boldsymbol{b'}]), & \text{otherwise} \end{cases}, \quad (54)$$

where $S_{no-island}$ is given by (48), $S_{island}$ by (52), and $z$ is given by (51) and evaluated using $a_{min}$ and $a'_{min}$ in (53).

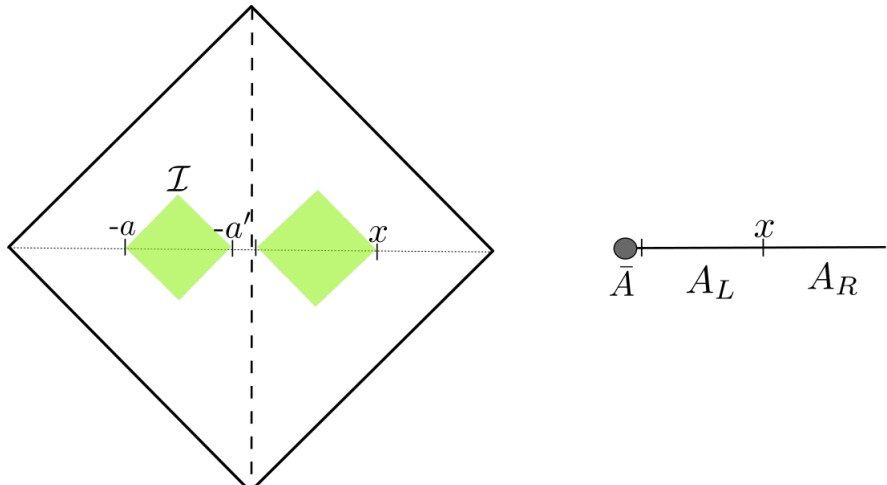

Figure 5: Left: Poincaré patch of AdS$_2$ coupled to a flat non-gravitating region. The green diamonds are the domains of dependence of $A_L$ and its island $\mathcal{I}$. Right: The holographic dual, a CFT on a half-line, with $A_L$ and $A_R$ the regions whose entropy we need in order to compute the entanglement contour $s_A(x)$.

We now apply this result to the intervals $A_L = (0, x]$ and $A_R = (x, \infty)$. For $A_L$, we have $b = 0$ and $b' = x$, which gives $a'_{min} = 6\phi_r/c$ and $a_{min} = a_{min}(x)$. We are only in the t-channel phase ($z > 1/2$) when

$$x > \frac{6\phi_r}{c}\left(1 + \frac{2}{\sqrt{3}}\right), \tag{55}$$

so there cannot be an island for $x$ smaller than this value. We make an additional assumption that $\phi_0 \gg c$, as then we also rule out $A_L$ having an island when $x$ of order $\phi_r/c$, because in this region $S_{island} \sim \phi_0$ while $S_{no-island} \sim c$. The last place to look for islands is $x \gg \phi_r/c$. Here we may approximate $a_{min}(x) \approx x$, and this gives us an island entropy

$$S_{island}(A_L) = 2\phi_0 + \frac{c}{6}\left(1 + \log\left(\frac{24\phi_r}{c\,\delta^2}x\right)\right) + O(c^0), \tag{56}$$

which is indeed less than $S_{no-island}(A_L)$ for all $x > x^*$, with this critical value $x^*$ approximately equal to

$$x^* = \frac{24\phi_r}{c}\exp\left(1 + \frac{12\phi_0}{c}\right). \tag{57}$$

This is the value of $x$ at which we switch from the no-island entropy to the island entropy:

$$S(A_L) = \begin{cases} \frac{c}{3}\log\left(\frac{x}{\delta}\right), & 0 < x < x^* \\ 2\phi_0 + \frac{c}{6}\left(1 + \log\left(\frac{24\phi_r}{c\,\delta^2}x\right)\right) + O(c^0), & x^* < x < \infty \end{cases}. \tag{58}$$

Now for $A_R$, where $b = x$ and $b' = \infty$. The no-island entropy of $A_R$ is IR divergent while, as we will see, the island entropy is IR-finite, so for $A_R$ we are always in the island phase for any value of $x$. From (52), we see that if $a$ were finite then the island entropy would also be IR divergent, but if $a = \infty$ then (52) is not the correct formula to use for island entropy; we should instead use (54) for the bulk entropy[7], which gives an IR finite result:

$$S(A_R) = 2\phi_0 + \frac{\phi_r}{a_{min}} + \frac{c}{6}\log\left(\frac{(a_{min} + x)^2}{a_{min}\delta^2}\right), \qquad 0 < x < \infty, \tag{59}$$

---

[7]The formula is not correct when $a = \infty$ for the same reason that $0 = S([-\infty, \infty]) \neq \lim_{L\to\infty} S([-L, L])$ for a pure state on a line.

with $a_{min}$ given by (53) with $x$ substituted for $b'$.

Now we can calulate the entanglement contour, dropping terms that are smaller than order $c$,

$$s_A(x)dx = \begin{cases} \frac{c}{24}\frac{1}{x^2}\left(x - 6\frac{\phi_r}{c} + \sqrt{x^2 + 36\frac{\phi_r}{c}x + 36\frac{\phi_r^2}{c^2}}\right)dx\,, & 0 < x \le x^* \\ 0\,, & x^* < x \end{cases}. \tag{60}$$

This is the main result of the subsection. This contour diverges like $c/x$ as $x \to 0^+$, due to the entanglement between UV degrees of freedom across the cut, and it monotonically decreases with increasing $x$ until the critical position $x^*$ where the contour discontinuously transitions from order $c$ to order 1[8]. For $x > x^*$, $S(A_L)$ is in its island phase and the order $c$ $x$-dependent terms of $S(A_L)$ and $S(A_R)$ are the same, so their difference is a constant. Except for its $c$-scaling, we do not know anything about the contour past $x^*$ because we do not know the subleading terms in the large-$c$ bulk entropy approximation (50). We now discuss physical implications of our result.

## 5.3 Physical implications of the bath entanglement contour

The most interesting feature of the bath entanglement contour we have calculated is its discontinuous transition from order $c$ to order 1 for all $x > x^*$. This is connected to quantum error correction and the ability to reconstruct states from their reduced density matrices. To show this we need a result from [45], which is that for any state $\rho_{ABC}$ there exists a quantum operation $\mathcal{T}_{B \to BC}$ such that one can approximately reconstruct the state from the reduced density matrix $\rho_{AB}$,

$$\sigma_{ABC} = \mathcal{T}_{B \to BC}(\rho_{AB})\,, \tag{61}$$

with a fidelity[9] to the true state $\rho_{ABC}$ of at least

$$F(\rho_{ABC}, \sigma_{ABC}) \ge 2^{-\frac{1}{2}I(A:C|B)_\rho}\,. \tag{62}$$

This means that the conditional mutual information quantifies just how precisely (as measured by the fidelity) $\rho_{ABC}$ can be reconstructed from its reduced density matrices $\rho_{AB}$ (or $\rho_{BC}$). One corollary is that if $I(A:C|B)_\rho = 0$, i.e. the state $\rho_{ABC}$ saturates the strong subadditivity inequality, then there exists a (state-dependent) recovery map which can reconstruct $\rho_{ABC}$ from $\rho_{AB}$ (or $\rho_{BC}$) with perfect fidelity. When $I(A:C|B)_\rho = 0$, the state on $A \cup B \cup C$ is a quantum Markov chain, meaning that there is no entanglement between $A$ and $C$, and no correlations except those mediated through $B$ [46]. The result (62) states that conditional mutual information quantifies how approximately Markovian a state is.

Let us apply these results to the system whose entanglement contour we calculated. In the non-gravitational half-line description, the regions of interest are the bath, $A = (0, \infty)$ and the 'black hole' degrees of freedom $\bar{A}$ at the end of the half-line. We now partition the bath into $A_L \cup A_i \cup A_R$, with $A_i$ an arbitrary subinterval $A_i := [b, b']$ of the bath. The partial entanglement entropy of $A_i$ by definition equals

$$s_A(A_i) = \int_b^{b'} dx\, s_A(x)\,, \tag{63}$$

with $s_A(x)$ the entanglement contour (60) we calculated earlier. The result (62) tells us that the smaller $s_A(A_i)$ is, the greater the accuracy with which $\rho_{\bar{A}A_L A_i}$ can be reconstructed from

---

[8]For the contour to still be order $c$ by the time $x$ reaches $x^*$, and so have a definite discontinuity, we have to make the additional assumption that $\exp(6\phi_0/c) \ll c$.

[9]$F(\rho, \sigma) := ||\sigma^{1/2}\rho^{1/2}||_1$ and from its definition $1 \ge F(\rho, \sigma) \ge 0$

its reduced density matrices $\rho_{\bar{A}A_L}$ and $\rho_{A_LA_i}$; there is guaranteed to exist a recovery map that reconstructs the state with a fidelity of at least

$$F(\rho_{\bar{A}A_LA_i}, \sigma_{\bar{A}A_LA_i}) \geq 2^{-s_A(A_i)}. \tag{64}$$

From the contour result we know that $s_A(A_i)$ is order 1 if $A_i \cap (0, x^\star) = \varnothing$, in which case we are guaranteed to be able to reconstruct the state with finite (i.e. not $\sim e^{-c}$) fidelity. When $A_i \cap (0, x^\star) \neq \varnothing$, the partial entanglement entropy is order $c$, in which case there is no such guarantee. These statements about the consequences of a vanishing entanglement contour are made purely within the non-gravitational description.

In the gravitational description, $s_A(A_i)$ is order 1 when $A_i \cap (0, x^\star) = \varnothing$ because the entanglement wedges of $\bar{A} \cup A_L \cup A_i = [0, b']$ and $A_L \cup A_i = (0, b']$ are identical, except for the small interval $[-6\phi_r/c, 0]$. This depends crucially on $A_L \cup A_i$ having a non-empty island region,

$$\mathcal{I} = [-b', -6\phi_r/c]. \tag{65}$$

In the language of entanglement wedge reconstruction, while the bulk region $[-b', -6\phi_r/c]$ is always protected against erasure of bath region $A_R$ whether or not $s_A(A_i)$ is order $c$, it is only when $s_A(A_i) \sim 1$ that it is also protected against erasure of $\bar{A}$, because then $\mathcal{I}$ is an island of $A_L \cup A_i$. So we see a connection between the CMI proposal, islands, and protection against erasures of the black hole degrees of freedom. To sum up, conditional mutual information quantifies how well a state can be reconstructed from certain reduced states, and this has a nice manifestation in island phase transitions where large bulk regions become protected from erasure.

## 6 Limitations of the CMI proposal

The CMI proposal for partial entanglement is only well-defined when elements in the partition of $A$ are totally ordered. The proposal is

$$s_A(A_i) = \frac{1}{2}(S(A_LA_i) - S(A_L) + S(A_RA_i) - S(A_R)). \tag{66}$$

Earlier we assumed elements in the partition are totally ordered, and defined $A_L$ and $A_R$ as $A_L := A_1...A_{i-1}$ and $A_R := A_{i+1}...A_n$. Where does this ordering come from?

When $A$ is an interval on a line, which is what we restricted ourselves to in applications, there is only one natural way of ordering those elements: from their order on the line[10]. The CMI proposal is well-defined in this class of examples.

As we will shortly explore with greater depth, when there is only one spatial dimension there is in general a natural and unambiguous order to elements in the partition, though with a few exceptions, such as when $A$ is the union of disjoint intervals. In higher dimensions however, there is no natural order to elements in the partition of $A$, though again with a few exceptions. In higher dimensions the CMI proposal is not well-defined, or to be more precise, the proposal is well-defined given a totally ordered partition of $A$, but that order is generally ambiguous and arbitrary in higher dimensions.

These difficulties do not undermine the demonstration that local measures of entanglement are useful tools for getting a sharp, spatially fine-grained picture of entanglement structure. The CMI proposal for entanglement contours has proven useful in 1+1d, but there is not yet a proposal that is generally applicable in higher dimensions. Our view is equivocal as to whether it is a refinement of the CMI proposal, or some other local measure of entanglement that is used in the future.

---

[10]Recall that numbering from left to right and right to left gives the same result.

$$\overline{A} \quad \quad A_2 \quad A_1 \quad \quad A_4 \quad \quad A_3 \quad \overline{A}$$

Figure 6: An example in 1+1d showing how giving the subdivisions an ordering that is different from the natural ordering can give partial entanglement entropies that do not have the expected symmetry properties.

## 6.1 Ordering ambiguities and symmetry in 1+1d

First we see what goes wrong in 1+1d if we do not order subdivisions of $A$ in the same order as they come on the line; the natural ordering. The entanglement contour of a single interval in 1+1d with a translation invariant state should have a $\mathbb{Z}_2$ symmetry about the interval's centre, and so too should the partial entanglement entropies of subdivisions that share that symmetry. The ordering of the subdivisions affects the symmetry properties, and even in 1+1d one can get partial entanglement entropies with unphysical symmetries if the natural ordering is not used. As an example, take the vacuum state of any field theory on $\mathbb{R}^{1,1}$, with an interval $A$ partitioned into four subdivisions, and label these subdivisions with the ordering shown in Fig. 6.

Take the lengths of intervals $A_2$ and $A_3$ to be the same. By symmetry we would expect the subdivisions labelled $A_2$ and $A_3$ to contribute equally to $S(A)$, and so their partial entanglement entropies should be equal, but it is easy to check from the CMI proposal formula (16) that this is not the case:

$$s_A(A_2) = \frac{1}{2}(S(A_1 A_2) - S(A_1) + S(A_2 A_3 A_4) - S(A_3 A_4)), \tag{67}$$

and

$$s_A(A_3) = \frac{1}{2}(S(A_1 A_2 A_3) - S(A_1 A_2) + S(A_3 A_4) - S(A_4)). \tag{68}$$

This issue is resolved in 1+1d by requiring natural ordering of the subdivisions of $A$ by the order they come on the line. It makes no difference if they are ordered left to right or right to left.

## 6.2 Disjoint intervals

The ordering of subdivisions in 1+1d is unambiguous when $A$ is a single interval, regardless of the space that that interval lives on. The ordering is also unambiguous when $A$ is a union of disjoint intervals, as long as those intervals all lie on a single line. However, we are in trouble when the intervals lie on a circle, or disconnected spaces. If for example $A$ is the union of two intervals on a circle, then it is not clear what to call the 'left' and 'right' of a subdivision of one of the intervals. In these cases, the CMI proposal is ambiguously defined, with dependence on the choice of ordering.

Note that we never need to worry about $A_i$ being a union of disjoint intervals, because by additivity property (II.) the partial entanglement entropy of such a region is the sum of partial entanglement entropies of its constituent intervals.

## 6.3 Ordering ambiguities in higher dimensions

In $d > 2$ there is no natural ordering to elements in a partition of $A$, and the CMI proposal for partial entanglement entropy is ambiguously defined. There are a few exceptions, such as when the way $A$ is partitioned is engineered to effectively reduce the dimensionality of the problem back down to 1+1d, as in [11].

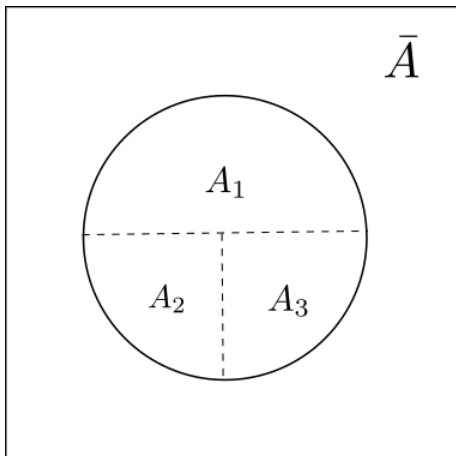

Figure 7: An example in 2+1d where the a choice of subdivision ordering gives partial entanglement entropies that do not share the symmetry of the state.

Worse still, the ordering ambiguity makes it impossible to even define the additivity property which the partial entanglement entropy is supposed to satisfy. Given a totally ordered partition of $A$, if we join two subdivisions $A_i$ and $A_j$ which share a spatial boundary together but with $|j - i| \neq 1$, where would $A_i \cup A_j$ come in the ordering of the new partition? This is not a problem in 1+1d, because subregions that are spatial neighbours are also neighbours in the order.

Lastly, certain orderings give partial entanglement entropies which lack the symmetry of the state and partitioning of $A$, and unlike 1+1d there is not a natural ordering to remedy that problem. As a basic example of how the subdivision ordering ambiguity can lead to symmetry problems, consider a 2+1d field theory on a spatial plane in the vacuum state, with a disk $A$ partitioned into three subregions, as shown in Fig. 7. From the symmetries of the setup we expect $s_A(A_2) = s_A(A_3)$, but again it is easy to check from the definition (66) that this is not the case. It would have had the expected symmetry if the ordering were such that the half-disk is $A_2$.

## 6.4 Resolving ordering ambiguities

We have explored the difficulties in defining the CMI proposal in higher dimensions. If we want to extend the domain of applicability of the proposal we either have to accept the ambiguities as an intrinsic part of the proposal, or try to resolve the ordering ambiguities.

In the spirit of accepting the ambiguities as an intrinsic part of the definition, we could imagine that, given an $n$ element partition of $A$, the CMI proposal gives us a different set of partial entanglement entropies for every possible way of ordering those elements. One could think of these orderings as $n!$ different ways of building up the entanglement between $A$ and $\bar{A}$. The ambiguities in what the partial entanglement entropies are are similar to the ambiguities of the boundary flux density of a flux maximising bit thread configuration, and it would interesting to find out whether there are any universal features of the conditional mutual information that are insensitive to the ordering of elements in the partition.

There are a few possibilities for trying to resolve ordering ambiguities in higher dimensions by modifying the CMI proposal. One way is simply to average over all possible orderings. This resolves the ambiguity, but the additivity property is still not well defined.

Another way to resolve the ordering ambiguity involves first grouping the elements in the partition of $A$ by their degree of separation from $\bar{A}$. An example is shown in Fig. 8. $A_1$ is the union of all elements that share a boundary with $\bar{A}$, and $A_{i+1}$ is the union of all elements

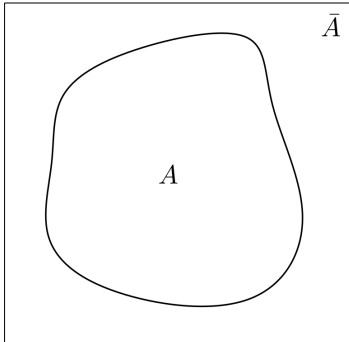
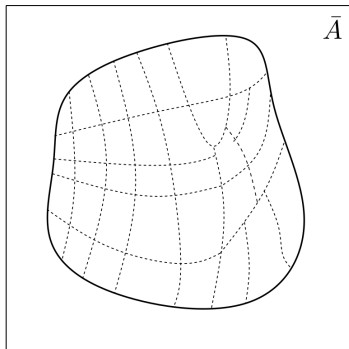
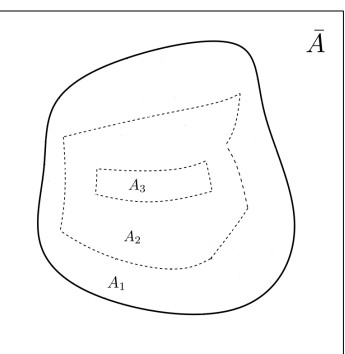

Figure 8: One procedure that resolves ordering ambiguity for the CMI proposal in higher dimensions. Elements in the partition of $A$ are sorted into equivalence classes that are defined and ordered by their elements' degree of separation from $\bar{A}$.

where you have to pass through at least $i$ subdivisions to reach $\bar{A}$. This grouping effectively reduces the problem down to 1+1d, where the CMI proposal is unambiguously defined. Back in section 3.2 we motivated the CMI proposal by building up entanglement between $A$ and $\bar{A}$ piece by piece, and the procedure described here can be thought of as a particular way of doing so in higher dimensions, by starting from the outer edge of $A$ and moving in.

This proposition is well-defined and unambiguous for any partitioning of $A$, but it also has several disadvantages: (1) the grouping of partition elements by degree of separation is a coarse-graining step which sacrifices some fine detail, (2) it can't in practice be evaluated for irregularly shaped regions, though this is a problem common to many entanglement measures, and (3) it does not give a well-defined entanglement contour, as the infinitesimal limit of partial entanglement entropy depends on the shape of the subregions as they are shrunk down.

Extending entanglement contours to higher dimensions may be possible with a modification of the CMI proposal, or it may require a new proposal perhaps based on a different set of physical requirements.

# 7 Future directions

There have been a couple of suggestions made in the paper for future research directions.

As discussed in section 6, the CMI proposal for partial entanglement entropy is well-defined in one and only one spatial dimension, with a few exceptions. One pressing task therefore is to find an entanglement contour formula that works in higher dimensions. It should satisfy a reasonable set of physical requirements, and be well-defined and unambiguous for any subregion $A$ and in any dimension. This may be a refinement of the CMI proposal, such as discussed in [47], or something entirely new. Expecting uniqueness of entanglement contour prescription may be too much, but at the very least the prescriptions allowed by the physical requirements chosen should yield contours with the same *qualitative* features. Otherwise, there is little physical meaning.

The CMI proposal for the entanglement contour in 1+1d, given by (17), involves a derivative in the spatial direction, and if a higher dimension refinement of the proposal is analogously defined in terms of shape deformations of entanglement entropy, then it would be interesting to connect it to previous work on the subject [48–51].

Entanglement contours of Hawking radiation provides a few interesting directions to pursue. In this paper we studied a static 2d model and found an entanglement contour that approximately vanishes except for an interval near the black hole degrees of freedom. We

discussed how this was intimately related to state reconstructibility and the appearance of islands. We do not know whether a vanishing contour is special to the particular proposal for PEE we used - the CMI proposal - or perhaps to the model used.

It would be interesting to calculate entanglement contours and other local measures of entanglement of radiation in *dynamic* models of black hole evaporation, where we can see the spatial entanglement structure of the Hawking radiation as it is purified after the Page time. The CMI proposal can be immediately applied to finite-temperature and dynamic 2d models with only calculational difficulties. Before the Page time, while the entanglement entropy of the radiation grows linearly, we would expect the contour to look something like $s_{rad}(x,t) \propto \Theta(t-x)$ as the early radiation enters the bath. After the Page time, when the late radiation starts purifying the early radiation, it is not obvious how the contour will look other than that the integrated contour $\int dx s_{rad}(x,t)$ must start decreasing.

One point worth stressing is that these 2d black hole models are rich yet tractable enough that we can calculate the von Neumann entropy of not only the bath, but of arbitrary subregions of the bath, and so we ought to be able to learn more about the entanglement structure than just the Page curve. This tractability has already been exploited for example in [52].

Lastly, von Neumann entropy is not the only quantity that one can define a contour function for. Contour functions have been defined for other quantum information quantities such as the entanglement of purification, and logarithmic negativity [7,53]. Other measures it may be interesting to find local notions of include mutual information and momentum space entanglement [54]. In holographic models of evaporation, islands are thought to be encoded in the radiation's large distance entanglement, which may have an interesting manifestation in a momentum space entanglement contour.

# Acknowledgments

We thank Matt Headrick and Qiang Wen for comments on an earlier draft, and Gurbir Arora, Jan de Boer, Harsha Hampapura, and Ian MacCormack for useful discussions. This research was supported by the research programme "Scanning New Horizons" (project number 16SNH02), which is financed by the Dutch Research Council (NWO), DoE grant DE-SC0009987 and the Simons Foundation.

# A    Properties of quantum conditional mutual information

The CMI proposal [4,7] for the partial entanglement entropy $s_A(A_i)$ can be understood as a conditional mutual information between $A_i$ and the purifier of $A$, $\bar{A}$, conditioned on either of the two intervals inbetween $A_i$ and $\bar{A}$,

$$s_A(A_i) = \frac{1}{2}I(A_i : \bar{A}|A_L) = \frac{1}{2}I(A_i : \bar{A}|A_R). \tag{69}$$

In this appendix we list properties of conditional mutual information in order to better understand this proposal and its relation to other quantum information quantities.

- **Definition in terms of von Neumann entropy**

$$I(A:B|C) = S(AC) + S(BC) - S(ABC) - S(C). \tag{70}$$

- **Symmetry**

$$I(A:B|C) = I(B:A|C). \tag{71}$$

- **Duality relation**

  For a four-party pure state on systems ABCD,

  $$I(A:B|C) = I(A:B|D). \tag{72}$$

- **Non-negativity**

  $$I(A:B|C) \geq 0. \tag{73}$$

  This inequality is equivalent to the strong subadditivity (SSA) relation of von Neumann entropies, which follows immediately from the definition (70).

- **Chain rule**

  $$I(A:BC|D) = I(A:C|D) + I(A:B|CD). \tag{74}$$

- **Upper bound**[11]

  $$I(A:B|C) \leq \min\{S(A), S(B), S(AC), S(BC)\}. \tag{75}$$

- **Pinsker-like lower bound** [55]

  $$I(A:B|C) \geq \frac{1}{4} \|(\rho_{ABC} - \exp(\log \rho_{AC} + \log \rho_{BC} - \log \rho_C))\|_1^2, \tag{76}$$

  where $\|A\|_1 := \text{Tr} \sqrt{A^\dagger A}$ is the trace norm.

- **Holographic bounds**

  In holographic theories dual to classical Einstein gravity, the conditional mutual information obeys the bounds

  $$I(A:B) \leq I(A:B|C) \leq 2E_W^G(AC:BC). \tag{77}$$

  These are bounds are tighter than non-negativity lower bound and the upper bound (75) satisfied in general, non-holographic theories. The lower bound follows from monogamy of mutual information [56], while the upper bound is in terms of a generalised notion of entanglement wedge cross section [57], defined as the minimal area surface that separates $A$ and $B$ in a region defined by differences in entanglement wedge regions

  $$E_W^G(AC:BC) := \min_\Gamma(\text{Area}(\Gamma)), \quad \Gamma \in \{\Gamma \,|\, \Gamma \subset r_{ABC} \backslash r_C \text{ and } \Gamma \text{ splits } A \text{ from } B\}, \tag{78}$$

  where $r_X$ is the entanglement wedge of boundary region $X$.

- **Relation to relative entropy** [58]

  $$\begin{aligned} I(A:B|C) &= S(\rho_{ABC} || \exp(\log \rho_{AC} + \log \rho_{BC} - \log \rho_C)) \\ &= S(\rho_{ABC} || \rho_A \otimes \rho_{BC}) - S(\rho_{AC} || \rho_A \otimes \rho_C). \end{aligned} \tag{79}$$

- **Markov chains**

  $I(A:C|B) = 0$ if and only if SSA is saturated. This is equivalent to $A - B - C$ being a quantum Markov chain [46,59], and the state on $ABC$ being [60]

  $$\rho_{ABC} = MM^\dagger, \tag{80}$$

  where

  $$M := (\rho_{AB}^{1/2} \otimes \mathbb{1}_C)(\mathbb{1}_A \otimes \rho_B^{-1/2} \otimes \mathbb{1}_C)(\mathbb{1}_A \otimes \rho_{BC}^{1/2}). \tag{81}$$

  In quantum Markov chains nothing is learned about the reduced state $\rho_A$ from learning $\rho_B$, given prior knowledge of $\rho_C$.

---

[11]For finite energy states in local field theories, where the von Neumann entropy of any region is UV divergent, this inquality is trivial.

- **Rényi conditional mutual information** [55]

Von Neumann entropy can be calculated from the analytic continuation of Rényi entropies, and conditional mutual information similarly has a Rényi generalisation

$$I(A:B|C) = \lim_{\alpha \to 1} \frac{1}{\alpha - 1} I_\alpha(A:B|C), \tag{82}$$

where

$$I_\alpha(A:B|C) := \log \mathrm{Tr} \left( \rho_{AC}^{(1-\alpha)/2} \rho_C^{(\alpha-1)/2} \rho_{BC}^{1-\alpha} \rho_C^{(\alpha-1)/2} \rho_{AC}^{(1-\alpha)/2} \rho_{ABC}^{\alpha} \right). \tag{83}$$

We have suppressed tensor products with the identity in this expression.

- **Relation to squashed entanglement**

The squashed entanglement of a bipartite state $\rho_{AB}$ is defined as the infimum of the conditional mutual information between $A$ and $B$, conditioned on possible tripartite extensions $\rho_{ABC}$,

$$E_{sq}(\rho_{AB}) := \inf_{\rho_{ABC}} \left\{ \frac{1}{2} I(A:B|C) : \rho_{AB} = \mathrm{Tr}_C \rho_{ABC} \right\}. \tag{84}$$

This implies another lower bound on the partial entanglement entropy:

$$s_A(A_i) \geq E_{sq}(\rho_{A_i \bar{A}}). \tag{85}$$

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
