# Peer review of "Local measures of entanglement in black holes and CFTs"

_SciPost Physics, doi:SciPost Phys. 12, 079 (2022)_

## Round 1 · Referee Report · Qiang Wen (Referee 1) · 2021-9-26

Strengths

1, Contains important and novel results on several aspects of entanglement contour and partial entanglement entropy;

2, Potentially inspires new research directions about entanglement contour and quantum gravity;

3, The paper is easy to understand, very clearly and carefully written;

Weaknesses

The discussion on resolving the ordering ambiguities in higher dimensions is not very convincing to me. As was mentioned by the author, this remains to be an open problem.

Report

The entanglement contour (or partial entanglement entropy (PEE)) quantifies how much different degrees of freedom (or sub-regions) within a given region A contribute to the entanglement entropy of A. The PEE is a newly proposed quantity in quantum information, which possesses the special property of additivity. The entanglement entropy is only a number hence hard to capture all the information in the density matrix. The entanglement contour is expected to give a finer description for the entanglement structure of quantum systems and furthermore, to help us better understand holography via quantum entanglement.

Although there have been several important progress in this direction, the study on PEE or entanglement contour is still on a primitive stage. The results in this paper has significantly expanded our understanding of many aspects of PEE. I am very happy to recommend this paper to be published in Scipost Physics in the present form.

I list the main results of this paper and my comments in the following:

1, Previously the PEE in 1+1 dimensions was proposed to be given by a linear combination of certain subset entanglement entropies. Here the author made an important observation that, this linear combination is indeed a conditional mutual information (CMI), hence relate the PEE to a known information theoretical quantity and help us better understand the physical meaning of PEE.

2, It is very interesting that the entanglement contour is used to explore the spatial structure of the Hawking radiation, which gives more information than the Page curve. Using the CMI proposal for PEE, the author showed that large bulk regions become protected from erasure in the island phase. This result is novel and inspires new research direction in quantum gravity. Also I list the typos I found

3, The paper also studied the entanglement contour for two dynamical system, the splitting quenches and the low energy excited states in 2d CFTs, and gave fine-grained picture of how the entanglement structure evolves in dynamical systems.

Requested changes

There may be a typo in the second line of (4.3), where ``$t \geq x$’’ should be ``$t \leq x$’’.

---

## Round 1 · Referee Report · Anonymous (Referee 2) · 2021-10-29

Strengths

1- The paper is well written, well formatted and easy to understand. 2- The paper cointains new results on entanglement entropy and applies them to cases of physical interest. Some interesting insights are uncovered.

Weaknesses

1- The generalization to higher dimensions seems not so straightforward. 2- It is still unclear how useful can a local measure of entanglement be, such as the one given by the specific entanglement contour proposal. The conditions are in principle physically motivated, however, entanglement entropy is intrinsically non-local. I wonder if these conditions are restricting the type of microstate in a non-trivial way.

Report

This paper explores in some detail a concrete proposal for partial entanglement entropy and entanglement contours, which are aimed to identify local contributions to entanglement entropy of a subsystem. The paper is well written and contains some novel results, some of which may be of interest to the community and inspire new avenues for future research.

Before recommending this article for publication, I have a couple of questions and possible ideas that could be implemented, which may help improve the overall quality of the manuscript.

1- Section 3: Can the formula (3.9) be applied to higher-dimensions if one consider the case of infinite strips (i.e. when one has translation invariance along the extra dimensions)? There are also known results for CFTs in the vacuum state and it would be interesting to see how the contours depend on the number of dimensions d, at least for cases with the given symmetry.

2- Section 3: I am intrigued about the connection between the entanglement contours based on CMI and bit threads. Section 3.5 deals with the relation of the CMI proposal to kinematic space, i.e., the space of geodesics, in the context of AdS_3/CFT_2. However, it is also well known that there are specific bit thread constructions based on geodesics, e.g., arXiv:1811.08879, which also work in higher dimensions. Can these constructions be used to define analogous "CMI contours" in higher dimensions?

3- Section 4: In 2d CFTs there are other well-known results for the modular Hamiltonian in dynamical cases, including global and local quenches. See, e.g., 1608.01283. I think these results can also be used to construct contours and might yield some interesting physical insights into the quasi-particle picture for entanglement propagation.

4- Section 5: is there any obstruction to consider an eternal two-sided black hole (at finite T) coupled to a bath, as in 1910.11077? I do not understand why one needs to restric to the zero temperature case. Note that from the two-sided case one can also extract a dynamical Page curve, which resembles the evolution of entanglement entropy after a global quench (linear growth and saturation at the Page time).

Requested changes

I would like the author to address points 1-4 in my report (either present explicit calculations or add relevant comments addressing these points). Once this is done I would be happy to recommend the article for publication in SciPost.

---

## Round 1 · Referee Report · Anonymous (Referee 3) · 2021-11-7

Strengths

1-The paper is very clearly written, with a careful and balanced introduction to the concept of entanglement contour and the recently developed technology of entanglement islands.

Weaknesses

1-In the particular context used, the entanglement contour reduces to a repackaging of the single interval entanglement entropies that have been derived in previous works, making it somewhat unclear whether genuinely new things are being found by this analysis --an exception to that, however, is Section 5.3, see main Report.

Report

The author explores the so-called entanglement contour, which is intended to be a measure of the ``density'' of entanglement entropy: How much a given subset of the degrees of freedom of a region contribute to its entanglement. They carefully review the ambiguities present in defining this notion and discuss the additional physical conditions that can be introduced to pin it down, in order to motivate a particular definition of the entanglement contour in terms of conditional mutual information (CMI). In the context of single interval entanglement contours in 1+1D QFT that the rest of the work focuses on, this reduces to a derivative of the Von Neumann entropy with respect to the interval's endpoints. In this sense, it is a repackaging of the information contained in the set of interval entropies.

They then explore the behavior of this object, first in quenched CFTs and in local excitations of the CFT vacuum, where they observe an interesting time evolution, and then in the context of 2D AdS black holes in JT gravity coupled to a 1+1D bath CFT in flat space. In the latter case, they see a reflection of the appearance of ``entanglement islands'' in the computation of bath subregion entropy in the corresponding entanglement contour: A jump of its value from O(c) to O(1) at the location where the island transition takes place. This is an interesting observation. It is perhaps somewhat unsurprising, given that we already knew that single interval entropy changes behavior at this point, however the relevant question is whether this behavior of CMI illuminates some new aspect of the island transition.
Section 5.3 attempts to derive such a consequence by pointing out the link between CMI and the fidelity of state reconstruction from its reduced density matrices. In particular, a connection is suggested between the appearance of an island in the entanglement wedge of a radiation interval and the protection of the global state against erasure of the black hole degrees of freedom. This is an intriguing observation, worth further investigation.

Overall, the author is knowledgeable of the subject and careful in their computations and arguments and the material of this work is of general interest for the community so I recommend publication.

---

## Round 2 · Author Response

I thank the referees for their careful reading and useful comments. I have added clarifications to the paper to address these comments.

---

## Round 2 · List of Changes

Report 2 Comment 1- Section 3: Can the formula (3.9) be applied to higher-dimensions if one consider the case of infinite strips (i.e. when one has translation invariance along the extra dimensions)? There are also known results for CFTs in the vacuum state and it would be interesting to see how the contours depend on the number of dimensions d, at least for cases with the given symmetry.

Clarification added to introduction: "In higher dimensions there are a few finely-tuned examples with sufficient symmetry, such as when A is a ball or infinite strip in the Minkowski plane, that the entanglement contour is effectively one-dimensional which makes the contour well-defined and calculable [5]. In the general non-symmetric higher dimensional case the formula (1.2) does not straightforwardly generalise."

Report 2 Comment 2- Section 3: I am intrigued about the connection between the entanglement contours based on CMI and bit threads. Section 3.5 deals with the relation of the CMI proposal to kinematic space, i.e., the space of geodesics, in the context of AdS_3/CFT_2. However, it is also well known that there are specific bit thread constructions based on geodesics, e.g., arXiv:1811.08879, which also work in higher dimensions. Can these constructions be used to define analogous "CMI contours" in higher dimensions?

Clarification added to introduction: " In examples where there exists a special set of bit threads based on geodesics [10] the boundary bit thread flux density has been calculated and shown to equal the entanglement contour calculated with the formula used in this paper [7, 11]."

Report 2 Comment 3- Section 4: In 2d CFTs there are other well-known results for the modular Hamiltonian in dynamical cases, including global and local quenches. See, e.g., 1608.01283. I think these results can also be used to construct contours and might yield some interesting physical insights into the quasi-particle picture for entanglement propagation.

Response: I agree that a natural and interesting direction for future work would be to consider the entanglement contours of perturbations about other states whose modular Hamltonian are known, which include quenched states with time-dependent modular Hamiltonians as the referee mentions.

Report 2 Comment 4- Section 5: is there any obstruction to consider an eternal two-sided black hole (at finite T) coupled to a bath, as in 1910.11077? I do not understand why one needs to restric to the zero temperature case. Note that from the two-sided case one can also extract a dynamical Page curve, which resembles the evolution of entanglement entropy after a global quench (linear growth and saturation at the Page time).

Clarification added to introduction: "We can calculate the entanglement contour of radiation in any 2d model of black hole evaporation, if the entanglement entropies of the intervals used in the contour formula are known. We explicitly calculate entanglement contour of a non-gravitational bath coupled to a zero temperature AdS2 black hole, which is one of the simplest setups where islands appear [24]. We find that the bath’s entanglement contour approximately vanishes everywhere except for a finite interval next to the ‘black hole’ degrees of freedom at the end of the half-line, and that this is caused by an island phase transition. This is because the entanglement contour quantifies how well the bath’s state can be reconstructed from its marginals, through its connection to conditional mutual information. Our calculation is only a first step in using local measures of entanglement to probe Hawking radiation, because the setup is static and so not a good model for black hole evaporation. We leave entanglement contours of finite temperature and dynamic black holes to future work."

Report 3 Comment 1-In the particular context used, the entanglement contour reduces to a repackaging of the single interval entanglement entropies that have been derived in previous works, making it somewhat unclear whether genuinely new things are being found by this analysis --an exception to that, however, is Section 5.3, see main Report.

Clarification added to introduction: "One objection to the formula (1.2) is that mathematically it is a mere repackaging of the entanglement entropy of various subregions, but the same objection can be levelled at quantities generally accepted to be useful such as mutual information and conditional entropy. The value is in the physical interpretation."

You are currently on this page

Resubmission 2107.11385v2 on 11 January 2022

---

## Editorial Decision

published